# Multi-Mechanistic Approaches to the Treatment of Traumatic Brain Injury: A Review

**DOI:** 10.3390/jcm12062179

**Published:** 2023-03-11

**Authors:** Daniel G. Lynch, Raj K. Narayan, Chunyan Li

**Affiliations:** 1Translational Brain Research Laboratory, The Feinstein Institutes for Medical Research, Manhasset, NY 11030, USA; 2Zucker School of Medicine at Hofstra/Northwell Health, Hempstead, NY 11549, USA; 3Department of Neurosurgery, St. Francis Hospital, Roslyn, NY 11576, USA; 4Department of Neurosurgery, Northwell Health, Manhasset, NY 11030, USA

**Keywords:** traumatic brain injury, combination therapy, multimodal therapy, multimodal neuromonitoring, pharmacologic, non-pharmacologic

## Abstract

Traumatic brain injury (TBI) is a leading cause of death and disability worldwide. Despite extensive research efforts, the majority of trialed monotherapies to date have failed to demonstrate significant benefit. It has been suggested that this is due to the complex pathophysiology of TBI, which may possibly be addressed by a combination of therapeutic interventions. In this article, we have reviewed combinations of different pharmacologic treatments, combinations of non-pharmacologic interventions, and combined pharmacologic and non-pharmacologic interventions for TBI. Both preclinical and clinical studies have been included. While promising results have been found in animal models, clinical trials of combination therapies have not yet shown clear benefit. This may possibly be due to their application without consideration of the evolving pathophysiology of TBI. Improvements of this paradigm may come from novel interventions guided by multimodal neuromonitoring and multimodal imaging techniques, as well as the application of multi-targeted non-pharmacologic and endogenous therapies. There also needs to be a greater representation of female subjects in preclinical and clinical studies.

## 1. Introduction

Traumatic brain injury (TBI) is a major global contributor to disability and death [1]. Central nervous system (CNS) damage induced by TBI is characterized by the initial primary injury with focal and diffuse tissue damage followed by secondary injury due to multiple pathophysiologic cascades, including neuroinflammation, ischemia, oxidative stress, excitotoxicity, and cerebral edema [1,2,3]. In spite of decades of fairly concerted efforts to minimize these secondary processes following TBI, there has been a history of promising results in preclinical studies that fail to show benefit in clinical trials [4]. To explain this failure, it has been suggested that TBI is not a homogenous disease state but rather a syndrome including a wide range of pathophysiologic derangements that undergo a complex and dynamic evolution over time [4]. Recognizing this, the National Institute of Neurologic Disorders and Stroke (NINDS) has since 2008 recommended research into combination therapies, including multiple different pharmacologic and/or non-pharmacologic interventions, to better address the multifactorial pathophysiology of TBI [5]. While initially promising, combination therapies for TBI have had mixed outcomes, with some studies demonstrating benefit and others failing to show significant benefit compared to individual treatments [6]. It has been proposed that failure to take into account the evolution of secondary injury mechanisms over time may have led to the mixed results seen in initial trials of combination therapies [7]. For example, the cellular and molecular drivers of oxidative stress are different in the initial minutes after TBI as compared to over the subsequent hours and days [8]. Researchers have suggested that an ideal treatment plan would not just administer a combination of therapies with different mechanistic targets, but also take into account how the evolution of secondary injury pathophysiology may play a different role at various time-points [7].

In the interests of clarity, it should be noted that following TBI, especially when severe, there are many secondary insults that may worsen the ultimate outcome, including systemic insults (hypotension, hypoxia, anemia, hyper/hypothermia, etc.) and intracranial insults (seizures, edema, vasospasm, hematoma expansion, etc.) [9]. In the past few decades, the proliferation of Neuro ICUs and Neuro Critical Care as a specialty acknowledges the recognition of these secondary insults and aims to prevent them, or to treat them expeditiously [10]. These secondary insults should not be confused with the secondary pathophysiological processes described earlier which constitute the focus of this review.

The need for integrated multitargeted treatments for TBI has been recognized [7], however, studies of such treatments are rare in the preclinical and clinical literature. The current state of multitarget treatment research has remained largely unchanged since NINDS first recommended investigation of combination therapies [5], consisting of combinations of treatments individually shown to be effective in prior single-target trials without necessarily considering the temporal evolution of TBI [6]. While such an approach has been proposed as a logical refinement of combination treatments [7], to date few if any trials have attempted this complex treatment protocol. However, there exists a large body of trials investigating combination therapies that, in conjunction with an understanding of the drivers of secondary injury, may allow for the creation of integrated multitarget treatment protocols. This review therefore seeks to summarize the progress that has been made on combination therapeutic interventions in the nearly 15 years since they were first recommended by NINDS, as well as highlight the utility of multimodal neuromonitoring and multimodal imaging to guide treatment regimens. In addition, current and ongoing work relating to multimodal treatment is discussed, with an emphasis on the integration of non-pharmacologic treatments and endogenous mechanisms.

## 2. Materials and Methods

### 2.1. Eligibility Criteria

A review of the available literature was conducted, selecting articles published up to December of 2022 (Figure 1). The inclusion criteria were: (1) Preclinical or clinical studies of TBI; including mild, moderate, and severe TBI in both adult and pediatric populations. (2) Studies investigating combinations of pharmacologic and/or non-pharmacologic interventions or investigating multiple multimodal neuromonitoring parameters used to guide treatment of TBI. Exclusion criteria included articles not published in English, abstracts, book chapters, and articles that could not be accessed for full-text review. Previous review articles were excluded from analysis, but studies cited within reviews were examined for eligibility.

### 2.2. Literature Search

The MedlinePlus and Cochrane databases were searched for relevant articles. Pharmacologic interventions were searched using “traumatic brain injury” AND “pharmacologic” OR “multimodal pharmacologic” OR “combination pharmacologic” (352 results). Articles investigating non-pharmacologic interventions were found with the keywords “traumatic brain injury” AND “non-pharmacologic treatment” OR “non-pharmacologic therapy” (259 results). The search string “traumatic brain injury” AND “combination therapy” OR “combination treatment” OR “multimodal treatment” OR “multimodal therapy” with keywords limited to title or abstract was used to identify any articles that may have escaped initial screening (2983 results). Articles investigating multimodal neuromonitoring as applied to TBI treatment were identified using “traumatic brain injury” AND “multimodal monitoring” OR “multimodal neuromonitoring” AND “treatment” (192 results). Articles were selected for inclusion based on title and abstract screening by an author (DGL).

### 2.3. Study Selection

After screening for articles that met inclusion criteria and adding articles missed on initial search, 209 potentially relevant articles on pharmacologic interventions and 170 articles investigating non-pharmacologic interventions were identified, along with 338 articles relating to multimodal neuromonitoring (MMM). After full text screening, 43 articles investigating combinations of pharmacologic interventions and 5 articles investigating non-pharmacologic interventions were selected for analysis. Additionally, 13 articles were identified as including combinations of pharmacologic and non-pharmacologic interventions and extracted into a separate category. After review, 12 relevant MMM-related articles were found that met selection criteria.

## 3. Results

### 3.1. Multi-Mechanistic Pharmacologic Interventions

TBI is a complex and heterogeneous disease process, and the secondary processes that follow TBI include oxidative stress, excitotoxicity, inflammation, neuronal loss, and glial cell activation. Given this wide variety of underlying drivers of secondary damage, multitargeted medications have been trialed to address the pathophysiology of TBI on multiple fronts [12]. However, these multitargeted interventions have to date largely been unsuccessful in clinical trials [5]. As the pathophysiology of secondary injury evolves over time, therapeutic interventions must be able to adapt to the evolution in molecular causes of injury; each medication is likely to have a unique therapeutic time window or windows based on the molecular timeline of secondary injury, during which it is most effective and outside of which it may lack significant benefit [13]. It should be noted that many interventions target secondary insults that occur after TBI (tissue ischemia, edema, hypotension, etc.) rather than the pathophysiologic mechanisms underlying these changes [2,14] which must be taken into account when comparing experimental protocols.

Following full-text review, 43 articles were identified as investigating a pharmacologic treatment strategy employing combinations of medications to treat preclinical or clinical TBI (Table 1). A diverse range of pharmacologic interventions were identified, including antioxidants, anti-convulsants, anti-excitatory, and anti-inflammatory compounds. Multiple interventions from different classes were typically applied in combination, to take advantage of synergistic mechanisms of action. The majority of trials (39/43) were in a preclinical animal model, with only three clinical studies.

#### 3.1.1. Antioxidant Treatments

Oxidative stress begins in the hyperacute phase, within seconds of TBI, due to disruptions in blood flow and cerebral metabolism that induce metabolic supply–demand mismatch in neurons, leading to activation of NADPH oxidase (NOX) [58]. NOX generates reactive oxygen species (ROS) and reactive nitrogen species (RNS) that contribute to DNA damage, mitochondrial dysfunction, and neuronal death, with the initial peak of NOX activity occurring in neurons approximately one hour after TBI, and a secondary microglia-mediated peak of NOX activity 24–96 h after TBI [58]. It should be noted that while decreased cerebral blood flow and tissue hypoxia have been seen in hyperacute TBI [59], this may not be associated with tissue ischemia but rather mitochondrial dysfunction, producing a paradoxical increase in ROS formation [60]. ROS and RNS also cause the peroxidation of lipids and proteins, disrupting the function of surviving cells and impairing the normal anti-oxidative response [8,61]. In a healthy brain, oxidative stress leads to induction of antioxidant enzymes, including superoxide dismutase (SOD), glutathione peroxidase (GPX), and catalase (CAT) [8], however, this response can be blunted in TBI [62]. Moving beyond the subacute period, oxidative stress and cell damage contributes to the chronic neuroinflammatory and neurodegenerative sequelae of TBI [8,61].

Antioxidant therapies, including free radical scavengers, antioxidant enzymes, and activators of endogenous antioxidant systems, have been trialed in TBI [8]. N-acetylcysteine (NAC) is a precursor to glutathione, an endogenous free radical scavenger that is known to be depleted in the subacute phase of TBI [8]. There is a lack of multimodal clinical trials of antioxidant therapies involving NAC, however, in a phase I randomized trial, NAC in combination with probenecid had no observed harmful effects, which may support future phase II and III clinical trials of similar antioxidant combinations [17]. L-arginine is a vasoactive amino acid which has been found to be depleted within hours of TBI, and low l-arginine levels are associated with increased RNS generation [63].

In preclinical models, a combination of arginine and antioxidant enzymes (CAT, SOD) has been used to modulate ROS generation and cerebral blood flow following TBI [55]. Some groups have begun to investigate novel multimodal pharmacologic interventions to better address the pathophysiology of TBI. A combination therapy including apocynin, tert-butylhydroquinone (tBHQ), and salubrinal was trialed in a preclinical model of TBI to target multifactorial causes of cellular stress [19]. Apocynin inhibits NOX, [64], while tBHQ upregulates genes encoding anti-oxidant enzymes [32,65] and salubrinal helps to reduce the production of misfolded proteins following TBI, preventing cell stress associated with accumulation of nonfunctional proteins [66], and has also been shown to reduce activation of pro-inflammatory cellular pathways [67]. This combination was found to improve lesion volume and improved functional outcomes after TBI, and this triple combination therapy had better outcomes compared to animals treated with only apocynin and tBHQ [19]. The authors also found these effects to be associated with reductions in activated microglia and peripheral immune cells in the peri-contusional cortex, along with decreased oxidative DNA damage. While these results must be replicated in large animal and clinical studies, they are promising for the future development of multi-mechanistic therapies.

#### 3.1.2. Anti-Excitatory Treatments

The processes underlying TBI-induced excitotoxicity begin within seconds of the initial trauma, with increased intracellular calcium ([Ca^2+^]_i_) from both increased cell membrane calcium permeability as well as release from intracellular stores [68]. This increase in [Ca^2+^]_i_ triggers release of vesicles containing neurotransmitters, including glutamate, into the synaptic cleft as early as 30 min following TBI [68]. Excess glutamate stimulates NMDA and AMPA receptors on nearby neurons, triggering depolarization and increased [Ca^2+^]_i_ that propagates a wave of excess excitation [69]. While this excitation is initially tolerable to neurons, mitochondrial failure begins to occur within four hours of trauma, leading to energy depletion, neuronal apoptosis, and neurodegeneration [68]. These alterations in [Ca^2+^]_i_ have been observed to persist up to seven days after TBI, in association with neuronal dysfunction [70]. Abnormal sprouting of collateral synapses, excitatory potentiation, and resulting hyperexcitability is observed in hippocampal neurons within days of TBI, increasing the risk for post-traumatic epilepsy [71].

Many medications are currently available to modulate cerebral excitation, several of which have been studied in TBI [72,73]. As part of a pharmacologic treatment protocol, combinations of anti-excitatory medications, including valproic acid, ketamine, perampanel, MK801, felbamate, and levetiracetam, have shown promise in reducing blood–brain barrier (BBB) disruption, expression of pro-inflammatory genes and cytokines, lesion volume, neuronal death, and functional deficits in preclinical TBI models [18,23,24,44,56]. Valproic acid inhibits sodium channels involved in excitatory neurotransmission, as well as increasing levels of the inhibitory neurotransmitter GABA in the CNS, making it useful in targeting the hyperacute post-synaptic excitation [74]. Levetiracetam inhibits the release of excitatory neurotransmitter-containing vesicles, which allows it to target the pre-synaptic neurons and intervene earlier to prevent excess excitation [74]. Ketamine, perampanel, MK801, and felbamate act at least in part through inhibition of NMDA and AMPA receptors, preventing the amplification of excitatory neurotransmission in the hyperacute and acute periods [74].

#### 3.1.3. Anti-Inflammatory Treatments

After TBI, lysis of cells as a result of the primary trauma or early secondary injury releases damage-associated molecular patterns (DAMPs), which are detected by toll-like receptors (TLRs) on nearby microglia [75]. The binding of DAMPs to TLRs activates the NF-kB/MAPK signaling pathway, driving secretion of pro-inflammatory cytokines, such as tumor necrosis factor alpha (TNF-α), interleukin 1-beta (IL-1β), and interleukin 6 (IL-6), with levels of these cytokines peaking within 1–2 days of the initial injury [75,76]. These cytokines activate nearby astrocytes and microglia, amplifying the pro-inflammatory signal and recruiting cerebral microglia and astrocytes as well as peripheral immune cells to the site of injury [76]. This neuroinflammatory response to TBI leads to neuronal and glial dysfunction, worsening secondary injury, and preventing repair of damaged cells [77]. Recruitment of local microglia is rapid, taking place within hours, while peripheral immune cells reach peak levels within the CNS 1–3 days following injury [76]. Microglia continue to be activated and recruited to the lesion for weeks to months following TBI [75,76,78]. Of note, microglia initially demonstrate an anti-inflammatory M2 phenotype peaking at one week post-injury, but at the 3–4 week point, a transition to the pro-inflammatory M1 phenotype is observed [75,76].

There have been several preclinical studies investigating combinations of anti-inflammatory medications to minimize neuroinflammation-associated secondary injury after TBI. Combinations of anti-inflammatory therapies, including doxycycline, minocycline, dexamethasone, and etanercept, have been trialed as part of a pharmacologic treatment approach and demonstrated reductions in pro-inflammatory cytokines, lesion volumes, BBB disruption, and neurobehavioral deficits, in addition to beneficial effects on glial cells and reductions in neuroinflammation [21,25,29,33,42,43,46,47]. Minocycline and doxycycline inhibit the mechanisms used by peripheral immune cells to enter the CNS, making them likely best applied in the late acute phase of TBI when neutrophils and macrophages are entering the lesion [79]. Etanercept decreases free TNF-α levels, which may reduce the burden of pro-inflammatory cytokines and decrease immune cell activation in the acute phase of TBI [77]. Dexamethasone and other corticosteroids act through inhibition of NF-kB/MAPK signaling, therefore preventing downstream secretion of cytokines, making them useful if applied immediately after the generation of DAMPs [77].

#### 3.1.4. Combined Multitarget Pharmacologic Therapies

While it is useful to discuss these mechanisms of secondary injury as separate and distinct processes, oxidative stress, excitotoxicity, and neuroinflammation are concurrent processes that overlap and interact after the brain injury. The infiltration and activation of immune cells in the acute period following TBI leads to the generation of ROS and RNS, further worsening oxidative stress [8]. Excitotoxicity and oxidative stress are also linked, as excessive excitatory stimulation and mitochondrial stress in the acute phase of TBI leads to the generation of ROS [80]. Furthermore, the NMDA and AMPA receptors involved in propagation of excitotoxicity are linked through the protein PSD95, which not only acts to amplify excitation early in the acute phase of TBI but also activates neuronal nitric oxide synthase, increasing the generation of RNS, with this relationship most apparent 24–48 h after TBI [69].

Several combinations of pharmacologic interventions have been tested, mostly in the laboratory, to target secondary damage following TBI. Combinations of antioxidant medications, including NAC and anti-inflammatory medications, have been shown to have beneficial effects on glial cells with reduced neuroinflammation and improved functional outcomes in preclinical studies [21,29,33,43]. Free radical scavenging antioxidant agents, such as lipoic acid and DHA in combination with anti-inflammatory medications including curcumin, have been used as an adjunct to neural stem cell grafting, with positive effects on cell graft survival and neuronal differentiation [20,31,39]. Combinations of NMDA receptor antagonist anti-excitatory medications, such as memantine and endogenous hormones including estrogen, have been shown to reduce neuronal death in preclinical models [34,38]. In clinical trials, the combination of the anti-excitatory GABA agonist propofol and the opioid fentanyl led to significant reductions in ICP, however, long-term outcomes were not reported [16].

### 3.2. Multimodal Nonpharmacologic Interventions

While the majority of the clinical and preclinical literature has examined pharmacologic interventions for TBI, nonpharmacologic interventions, including neuromodulation, lifestyle modification, physical exercise, and nutraceuticals, have been trialed in clinical and preclinical studies of TBI [81,82,83]. A treatment approach using combined nonpharmacologic interventions has a similar potential as pharmacologic interventions to address the evolving pathophysiology of TBI-induced secondary injury, modulating different molecular targets at different time points.

After full-text review, five studies investigating combinations of nonpharmacologic interventions were included for analysis, including two clinical trials and three preclinical studies (Table 2). While a wide range of nonpharmacologic treatments have been tested in the preclinical and clinical literature, the modalities most often investigated in combination with other treatments were transcranial magnetic stimulation (TMS) and environmental enrichment (EE).

#### 3.2.1. Clinical Trials

TMS is a noninvasive form of neuromodulation that has been widely applied in TBI due to promising initial results in neuroprotection and recovery in preclinical models [89,90]. While the molecular mechanisms underlying the effects of TMS are not fully understood, in models of cerebral ischemia, TMS has shown the ability to modulate neuroinflammation through inhibition of NF-kB and promotion of microglia M2 polarization [91], as well as antioxidant effects through upregulation of antioxidant enzyme transcription factors [92], making TMS theoretically a useful intervention early in the acute phase of TBI. TMS has been extensively applied as a monomodal therapy but has also been used in two clinical trials as part of a combination. The first clinical trial investigated the use of TMS and intensive neuromotor training in a patient with chronic neurologic deficits following severe TBI [85]. In this report, the multimodal therapy resulted in improved motor recovery, balance, and walking ability. A larger prospective trial investigated combined therapy with TMS and neurocognitive rehabilitation in patients after mild to moderate TBI, with improvement in neurologic and functional outcomes versus rehabilitation alone [84]. The same paper also found changes in cerebral metabolite ratios using MRS suggestive of improved neuronal energy metabolism in the group receiving combined therapy. However, it is important to note that both clinical trials implemented TMS as part of a long-term rehabilitation protocol weeks to months following the primary injury. At this “chronic” stage of TBI, the pathophysiological processes may have little resemblance to the acute phase. To date, no clinical trials have reported outcomes of combined nonpharmacologic interventions in the acute phase of TBI.

#### 3.2.2. Preclinical Trials

EE is an environment that provides enhanced cognitive, physical, and social stimulation, which is thought to help prevent deterioration following TBI [93]. While well-studied as a single intervention, several preclinical studies have examined EE as part of a combined treatment approach in which it reduced neuronal apoptosis and astrocyte activation with resulting improvements in neurocognitive dysfunction [87,88]. As part of a multimodal treatment paradigm including EE, TMS was also found to improve motor and sensory function after TBI [86].

### 3.3. Multimodal Combined Pharmacologic and Nonpharmacologic Interventions

Following full-text review, 13 articles were identified as investigating a combination of pharmacologic and nonpharmacologic interventions following TBI, all but one of which were studied in the context of preclinical trials (Table 3).

EE has been used as part of a multimodal treatment protocol with the anxiolytic buspirone or the dopamine agonists amantadine or galantamine [94,95,96,97]. In the majority of cases, EE and the pharmacologic intervention were each beneficial but had no additional benefit when combined. In one study, however, multimodal treatment with EE and buspirone improved functional outcomes in comparison to the individual treatments [96]. Several other pharmacologic and nonpharmacologic modalities have been explored in combination for the prevention of secondary injury following TBI. Therapeutic hypothermia and the growth factor FGF2 were each individually beneficial in improving lesion volume and resulting neurobehavioral outcomes but had no additional benefit when combined [98]. Other studies involving therapeutic hypothermia have been mixed, with a benefit versus monomodal therapies when combined with stem cells in preclinical studies [99] but worse outcomes when combined with progesterone in a large clinical trial [100]. Voluntary physical exercise, in combination with the endogenous compound citicoline, demonstrated reductions in lesion volumes and neurobehavioral outcomes after TBI, but there was no benefit to combined therapy compared to the individual interventions [101]. Lastly, a more recent study investigated the combination of mesenchymal stem cell (MSC) grafting with low-intensity transcranial ultrasound (LITUS) [102]. The authors found that MSC grafting improved lesion volume and neurobehavioral outcomes, and this could be significantly improved with the application of LITUS. Furthermore, the authors found that this effect was mediated in part via induction of brain-derived neurotrophic factor (BDNF) and reductions in TNF-α and aquaporin-4 expression.

**Table 3 jcm-12-02179-t003:** Combinations of pharmacologic and non-pharmacologic interventions studied in preclinical models of TBI.

Model	Species	Sample Size	%Fem	Intervention	Outcome	Reference
Clinical Trials						
Randomized Placebo Controlled Trial	Human Severe TBI	107	15.9%	“Progesterone + Hypothermia” vs. progesterone or hypothermia alone	Worse long-term outcomes in combined group vs. individual therapies in acute TBI	Sinha, 2017 [100]
Preclinical Trials						
rmCCI	Rat	40	0%	“Amantadine + tDCS” vs. amantadine or tDCS alone	Improved neurobehavioral outcomes, decreased astrocyte activation	Han, 2022 [103]
CCI	Rat	90	0%	“MSC + LITUS” vs. MSC or LITUS alone	Improved lesion volume and neurobehavioral outcomes, mediated through induction of BDNF and reduction of TNF-α and AQP4	Yao, 2022 [102]
CCI	Rat	68	0%	“Citalopram + EE” vs. citalopram or EE alone	Improved learning and cognitive flexibility	Minchew, 2021 [104]
FPI	Rat	96	0%	“BMSC + Hypothermia” vs. BMSC or hypothermia alone	Decreased neuronal apoptosis and neurobehavioral defects	Song, 2020 [99]
CCI	Rat	60	0%	“Amantadine + EE” vs. amantadine or EE alone	Improved lesion volume and neurobehavioral outcomes, no additional benefit versus monotherapy	Bleimeister, 2019 [94]
CCI	Rat	72	0%	“Galantamine + EE” vs. galantamine or EE alone	Improved lesion volume and neurobehavioral outcomes, no additional benefit versus monotherapy	de la Tremblaye, 2017 [95]
CCI	Rat	48	0%	“Methylphenidate + EE” vs. methylphenidate or EE alone	Improved neurobehavioral outcomes, no additional benefit versus monotherapy	Leary, 2017 [105]
CCI	Rat	48	0%	“Citicoline + exercise” vs. citicoline or physical exercise alone	Improved lesion volume and neurobehavioral outcomes, no additional benefit versus monotherapy	Jacotte-Simancas, 2015 [101]
CCI	Rat	78	0%	“Buspirone + EE” vs. EE or buspirone alone	Improved functional outcomes	Monaco, 2014 [96]
CCI	Rat	60	0%	“Buspirone + EE” vs. EE or buspirone alone	Improved functional outcomes, no additional benefit versus monotherapy	Kline, 2012 [97]
CCI	Rat	65	0%	“8-OH-DPAT + EE” vs. EE or 8-OH-DPAT alone	Decreased neuronal loss, no additional benefit versus monotherapy	Kline, 2010 [106]
CCI	Rat	50	0%	“FGF-2 + Hypothermia” vs. Hypothermia or FGF-2 alone	Improved lesion volume and neurobehavioral outcomes, no additional benefit versus monotherapy	Yan, 2000 [98]

AQP4, aquaporin-4; BDNF, brain-derived neurotrophic factor; BMSC, bone marrow derived mesenchymal stem cells; EE, environmental enrichment; FGF-2, fibroblast growth factor 2; LITUS, low-intensity transcranial ultrasound; MSC, mesenchymal stem cells; 8-OH-DPAT, 8-hydroxyl-2-(di-N-propylamino)tetralin; tDCS, transcranial direct current stimulation; TNF-α, tumor necrosis factor alpha.

### 3.4. Multimodal Neuromonitoring

Multimodal neuromonitoring (MMM) entails the measurement and integration of multiple biological parameters to better understand a patient’s pathophysiologic state and guide treatment decisions [107]. Measurement of cerebral physical, metabolic, and electrical physiologic parameters through MMM sensors allows for the collection of a wide range of physiologic and pathophysiological data. When interpreted and analyzed, insights gained from MMM can help determine what interventions, if any, are needed to optimize cerebral physiology [13]. The rationale for MMM is thus to convert the heterogenous pathophysiology of TBI into an array of distinct physiologic variables that may be more or less amenable to different forms of treatment at different time periods.

After full-text review, 12 articles were included for analysis (Table 4). Of note, while MMM has been extensively studied, articles were only included if they examined outcomes resulting from a combination of two or more different biological parameters in the context of MMM-guided treatment of TBI. Studies examining measurement parameters and implementation strategies alone, as well as studies not reporting outcomes or use of MMM to guide treatment were excluded. The most common physical parameters measured alongside ICP were brain tissue oxygenation (P_bt_O_2_), cerebral perfusion pressure (CCP), and sometimes cerebrovascular pressure reactivity (PRx). Metabolic brain parameters including brain tissue pH and lactate-pyruvate ratio (LPR) can be measured using cerebral microdialysis (CMD) and jugular venous oxygen saturation (S_jv_O_2_) measurement can be used to estimate cerebral oxygen extraction capability.

#### Multimodal Neuromonitoring-Guided Treatment

Among the most commonly studied MMM parameters are intracranial pressure (ICP), cerebral perfusion pressure (CPP), and cerebrovascular autoregulatory capability (PRx). As elevated ICP reduces CPP and prevents optimal brain perfusion, evidence-based guidelines recommend ICP monitoring for patients with severe TBI [10,107,120], as well as establishing ICP and CPP-based thresholds for pharmacologic and surgical interventions [10]. TBI treatment based on MMM-derived optimal cerebral perfusion has demonstrated improved clinical outcomes, especially in older patients who may have reduced autoregulatory capacity [114]. PRx allows for real-time measurement of cerebrovascular autoregulation [121] and may help to predict the need for surgical intervention in severe TBI [112,122,123]. Measurement of brain oxygenation parameters, such as P_bt_O_2_, allows for rapid implementation of therapeutic interventions to correct cerebral hypoxia [109,121], and measurement of P_bt_O_2_ and ICP in the BOOST-II trial was suggested to improve secondary injury after TBI demonstrating the potential effect of such early interventions [116]. Measurement of P_bt_O_2_, CPP and arterial oxygen saturation has been used to guide treatment of cerebral hypoxia in severe TBI, helping to uncover physiologic states that may lead to impaired brain oxygen metabolism [115]. CMD-based parameters, including LPR and pH, can directly characterize cerebral metabolic states, including ischemia, metabolic stress, and mitochondrial dysfunction [107,124], and has been shown to predict cerebral metabolic crisis and disruptions in cerebral perfusion requiring therapeutic intervention [110,119]. The identification of metabolic derangement, including mitochondrial dysfunction via CMD-based MMM, has also been used to guide targeted metabolic treatment of TBI [111].

## 4. Discussion

Despite many decades of concerted effort, a highly effective treatment for TBI remains elusive [125]. Considering the multifactorial pathophysiology of TBI, a treatment paradigm combining interventions targeting more than one aspect of TBI pathophysiology has been suggested as the goal for ongoing and future research [5]. However, to date these combination therapies have demonstrated mixed results [6].

The available literature suggests that the reason for this lack of success may be that these combination therapies have not addressed the evolution of TBI pathophysiology over time. In recent years, the scientific understanding of pathophysiology driving secondary injury after TBI has grown immensely [2,3,14,69]. It is now well established that the mechanisms underlying key drivers of secondary injury, such as inflammation and oxidative stress, are not static but undergo change over time [8,68,126]. With this understanding, it may be possible to predict what cellular and molecular mechanisms are most likely contributing to secondary injury at a particular stage of injury, however, it is known that patients show a high degree of variability in their pathophysiologic response following TBI [127]. Therefore, combined treatments must be based on not only a detailed understanding of “typical” secondary injury evolution but also refined and fine-tuned in accordance with the patient’s individual development of secondary injury to best treat TBI. It is possible that a multimodal treatment paradigm may address the evolving pathophysiology of secondary injury through judicious application of pharmacologic and/or nonpharmacologic interventions that are informed by multimodal imaging and neuromonitoring. By addressing not only the multifactorial causes of secondary injury, but also how they change as the disease progresses, multimodal treatment may have the potential to succeed where combination therapies failed.

### 4.1. Current State of Multimodal TBI Treatment

While it is increasingly appreciated that clinical TBI is a heterogeneous group of disease processes rather than a single disease, the temporal evolution of TBI-induced secondary injury has been less well investigated in the setting of TBI treatment [7], which may contribute to the current gap in outcomes between preclinical and clinical studies. The primary drivers of TBI-associated secondary injury can be broadly divided into the categories of neuroinflammation, oxidative stress, and excitotoxicity, although many other mechanisms, including mitochondrial failure and cerebral edema, may also play a role. Each of these is not a static disease state, but rather an evolving disease process. For example, in the hyperacute state, neuroinflammation in TBI is driven by DAMP-induced activation of local brain tissue microglia, leading to secretion of pro-inflammatory cytokines within hours of the injury occurring [75]. However, in the acute period within days of the injury, infiltration of peripheral immune cells contributes more to neuroinflammation, and at the 3–4-week time point, pro-inflammatory M1 microglial activation becomes a prominent driver of the inflammatory process [76]. Given this, it is possible that, for example, an intervention targeting polarization of microglia from the M1 to the anti-inflammatory M2 phenotype may be largely ineffective if given in the acute phase but may show benefit in the early chronic phase of secondary injury. Additionally, an increased understanding of the molecular mechanisms driving secondary injury may help to stimulate novel treatments for TBI. For example, ferroptosis is a form of iron-mediated regulated cell death that is increasingly understood to mediate much of the damage associated with oxidative stress in TBI [80]. Iron chelators, such as deferoxamine, have been shown to mitigate ferroptosis following TBI [128], with resulting beneficial effects on TBI-associated edema, hydrocephalus, and neurotoxicity [129,130,131]. While inhibition of ferroptosis has had promising results as a monotherapy, combination and multi-mechanistic approaches targeting this mechanism have yet to be reported.

Additionally, treatment for secondary injury after TBI is most commonly directed at stopping or preventing secondary insults, such as hypotension or cerebral edema, that can occur in the absence of careful management [9,10,14,132]. Treatment approaches focusing on the underlying physiological changes of secondary injury, such as excitotoxicity and neuroinflammation, are less commonly reported [133,134]. It is thus important to not conflate secondary insults with the pathophysiology of secondary injury in comparing treatment paradigms.

Many pharmacologic interventions have been trialed to prevent or reduce secondary injury resulting from TBI. In preclinical models, an extensive number of medications have been tested to prevent secondary injury, and nearly all tested combinations have demonstrated some degree of improvement in neurobehavioral outcomes and lesion volume after experimental TBI (Table 3). However, none have been clinically successful to date as a monotherapy or combination therapy [5,6]. A multimodal treatment approach taking into account the evolving pathophysiology of TBI may be the key for clinically successful treatments of secondary injury. While there are some promising preclinical trials testing a similar method [19], there have been no clinical trials to date investigating this multimodal treatment strategy. While there are many potential options for targeted multimodal treatments, one option for pre-screening medications likely to be effective is to trial medications that showed promise in preclinical and early clinical (phase I-II) trials but did not show a benefit in phase III clinical trials. These medications have an established safety profile and proven mechanism, and it is possible that some of them may have had potential to be effective but were not applied in a multimodal manner, at the optimal time point based on the evolution of TBI pathophysiology and the patients’ physiologic state.

While the pathophysiologic mechanisms driving secondary injury have been better characterized in recent years, much of the research uncovering these mechanisms has been performed in male animal models or male patients [135]. For example, in the preclinical trials examined within this paper, only three out of fifty-four studies (6%) included female animals, and the average percentage of female subjects within analyzed clinical studies was 24%. The pathophysiologic mechanisms underlying primary and secondary injury following TBI are likely more similar than different males and females, however, some differences are known to exist between males and females in TBI. In preclinical models of TBI, female animals have demonstrated decreased neuroinflammation, including reductions in reactive microglia and infiltrating peripheral immune cells [136,137], decreased BBB disruption [138], and decreased oxidative stress [139]. While the mechanistic differences in TBI pathophysiology are complex and still under investigation, the effect of female sex steroids, including estrogen and progesterone, has been suggested as a key underlying driver for these observed differences [140]. In animal models, female sex hormones, including estrogen and progesterone, have shown efficacy as part of monomodal and combined therapies [38,40,52]. However, clinical trials of estrogen and progesterone as therapies for TBI have not been widely successful, which has been suggested to result from failure to account for physiologic differences in sex hormones due to age and biologic sex at the point of treatment [140]. The discrepancy between preclinical studies, in which female animals typically experience better outcomes, and clinical trials, in which females typically experience worse outcomes [135], additionally highlights the need for better understanding of sex differences in TBI pathophysiology and the need to consider sex hormonal levels as an important physiologic parameter that could help guide multimodal treatment.

### 4.2. Future Directions in Multimodal TBI Treatment

While the prevention of secondary injury following TBI remains a challenge for clinicians, there are several promising avenues for multimodal TBI interventions that are undergoing active research.

As the temporal evolution of TBI-induced secondary injury has become increasingly well understood, so too has grown understanding of the spatial differences in TBI pathophysiology and how this may guide treatment decisions. In TBI, there exist regions of contused brain tissue injured by direct trauma, surrounded by areas of peri-contusional tissue at risk for secondary injury [141,142]. While the contused areas experience severe metabolic and mitochondrial dysfunction that can culminate in unregulated cellular necrosis, the peri-contusional tissue typically maintains sufficient mitochondrial and metabolic function to initiate cellular repair processes [133]. This is a critical distinction, as therapeutics that rely on specific cellular pathways may be viable in the peri-contusional tissue but less so in the contused tissue. For example, glucocorticoids have the potential to reduce edema and inflammation following TBI, but act through activation of transcription factors that induce various downstream effector proteins [143]. In a large randomized clinical trial, administration of glucocorticoids was associated with an increased risk of death compared to placebo, which may result from this mechanistic failure and increased demand on already stressed cells [144]. A better understanding of the relative distribution of contused and peri-contused tissue in an individual patient may offer insights into the efficacy of a particular treatment or combination of interventions, however, this approach has not yet been tested in clinical studies.

Non-pharmacologic interventions offer some promise in the multimodal treatment of secondary injury after TBI, for several reasons. Many non-pharmacologic interventions, such as neuromodulation and nutraceuticals, can be initiated without disrupting ongoing treatment and have a favorable risk/benefit profile [81,82], which makes it easier to apply them alongside other therapies to address TBI as it evolves. However, much like for pharmacologic interventions, the blind application of non-pharmacologic treatment techniques without a detailed understanding of the patient’s physiologic state at the time of application, as well as knowledge of the molecular targets of the intervention, is likely to fail. As an example, TMS has been shown in ischemic brain tissue to work in part through induction of anti-oxidative enzymes [92] and inhibition of NF-kB [91], suggesting that TMS may be most effective if applied as part of a multimodal treatment regimen in the acute phase of TBI. Just as multimodal imaging and neuromonitoring have shown potential to guide pharmacologic therapies for TBI, they have also helped inform non-pharmacologic treatments. Several groups have used MMM parameters, including ICP, CPP, TCD, and P_bt_O_2_, to guide the use of therapeutic hypothermia in patients with severe TBI [145,146]. Additionally, applying a combination of pharmacologic and non-pharmacologic interventions is a possible means to address the pathophysiology of TBI and TBI-associated secondary damage, as non-pharmacologic interventions can target similar mechanisms of secondary injury through different targets in a multimodal approach to TBI treatment. For example, a hypothetical combination of mild therapeutic hypothermia and a free radical scavenger would target oxidative stress through decreasing ROS/RNS and upregulating anti-oxidative enzymes, potentially impacting oxidative stress at the hyperacute and early acute time periods [61,147]. While non-pharmacologic interventions are an appealing avenue for treatment of TBI, the mechanisms that underlie these interventions are not always well characterized. When the cellular and molecular targets of non-pharmacologic interventions for TBI are better understood, they may be a powerful tool for multimodal treatment of TBI.

Harnessing endogenous mechanisms with an inherent capability to activate multiple cellular pathways may allow for the targeting of TBI-induced secondary injury at distinct time points, which serves as the foundation for investigation of endogenous mechanisms as part of a multimodal treatment paradigm. Implemented in this fashion, it is possible that endogenous mechanisms may mimic the effects seen with administration of multiple pharmacologic treatments, potentially without the associated logistical challenges. Conditioning is a widely applied therapeutic technique in which a potentially harmful stimuli is applied below the threshold for tissue damage, leading to the induction of endogenous neuroprotective pathways [148]. Remote ischemic post-conditioning is a conditioning mechanism widely studied in ischemic stroke and found to be safe and effective in both preclinical and clinical studies [149,150], and has demonstrated promising results in TBI [151,152]. While the mechanistic effects underlying ischemic post-conditioning are still being uncovered, it is known to increase expression of BDNF and promote neurogenesis and neuroplasticity, which may help to prevent neurodegeneration and maladaptive functional alterations following cerebral insults [148]. Ischemic post-conditioning may thus be most effective if applied in the subacute and early chronic phases, to promote neuronal repair and functional recovery following TBI. The beneficial effects of conditioning-activated endogenous pathways parallel the investigation of therapeutic hypothermia for TBI, developed out of the observation that hibernating animals display resistance against TBI [153,154]. While preclinical trials have shown promise for therapeutic hypothermia, the available clinical literature does not demonstrate a mortality benefit from therapeutic hypothermia in adults [155,156]. However, this may act to reiterate the point that, with a disease as complex as TBI, it is necessary to apply a treatment at exactly the right point in disease progression, thus necessitating the implementation of therapeutic hypothermia as part of a multimodal treatment plan, including multimodal imaging and monitoring-based assessment of individual physiology. In fact, recent work has investigated the optimal timing of therapeutic hypothermia in preclinical studies [157], and a clinical trial has used CMD to guide therapeutic hypothermia with a resulting reduction in mortality [158], demonstrating the feasibility of such an approach. Additionally, a hibernation-like state can be induced using medications including psychotropics, which may allow for the application of therapeutic hypothermia via modulation of endogenous thermoregulation and reduce the logistical challenges associated with external cooling [159]. Though not yet assessed in TBI, the diving reflex is yet another endogenous mechanism that may be harnessed. Triggered when trigeminal sensory afferents are activated, such as by cold water during diving, a constellation of mechanisms, most notably driving blood away from the periphery and towards the brain, are produced [160]. Far from only increasing the flow of blood to the brain, activation of the diving reflex initiates the development of an anti-oxidative phenotype, both reducing the level of systemic ROS [161] and increasing antioxidant enzyme activity levels [162,163]. A systemic anti-inflammatory state is seen with chronic activation of the diving reflex [164,165], which may be due to the effects of the diving reflex being mediated in part through the vagus nerve, well known for its modulation of the inflammatory complex [166,167]. If applied judiciously, as part of a multimodal treatment strategy, the diving reflex thus could be used to target both the acute and subacute stages in oxidative stress development. Given these factors, it is possible that the diving reflex may be able to target multiple points in the timeline of TBI’s pathogenic progression and represents a promising avenue for future research. However, natural stimulation of the diving reflex has yet to be adopted for routine treatment of TBI for various reasons, including the limited currently available methods of induction (cold water facial cooling), its yet-to-be-defined dose–response, variable reproducibility, and challenges of implementing the diving reflex in the clinical setting [168]. Electrical stimulation of the trigeminal nerve has demonstrated the ability to induce the endogenous diving reflex in a reproducible, dose-controlled manner and thus may represent a more practical application of this neuroprotective endogenous mechanism in TBI [168,169,170]. The ability of endogenous mechanisms to modulate cellular pathways is attractive in the treatment of TBI.

### 4.3. Imaging- and Neuromonitoring-Guided Treatment

Multimodal imaging represents a wide array of imaging techniques, including MRI and PET, that are able to integrate structural and functional information in TBI, correlating regions of abnormal anatomy with disruptions in CNS function [171]. This is particularly important for mild TBI (mTBI), in which objective diagnosis and initiation of treatment is sometimes only possible with multimodal imaging techniques [172,173,174]. In moderate and severe TBI, multimodal imaging historically plays a greater role in prognostication and determining the need and appropriateness of aggressive interventions [175,176]. MRI has been studied in clinical trials, predominantly in mTBI in the subacute or chronic phase, as it is available in most academic medical centers and does not expose the patient to ionizing radiation [174,177]. MRI sequences, including diffusion weighted imaging (DWI), can be used to image cerebral edema following TBI, even in the first few days following injury [178,179], which could guide the early administration of therapies to reduce edema [14]. Other advanced MRI techniques, such as functional MRI (fMRI) and magnetic resonance spectroscopy (MRS), may have a role in guiding treatment decisions after TBI. fMRI measures changes in blood oxygen levels associated with neuronal activation to track brain activity in real time [180], and can localize foci of post-TBI epilepsy even in the absence of visible structural lesions [181], which may allow for early initiation of antiepileptic medications in high risk patients in conjunction with other clinical data [182]. MRS is a non-invasive means to quantify relative ratios of cerebral metabolites in specific brain regions [183], and has been used to assess neuroinflammation in preclinical models of TBI [184,185]. It has been proposed that a noninvasive means to detect neuroinflammation, including MRS and novel PET radiotracers specific to activated glial cells, could be used to guide optimal timing of anti-inflammatory therapies in TBI [186]. In this way, multimodal imaging techniques may be able to evaluate aspects of TBI-induced secondary injury at the micro-scale and fine-tune treatment. For example, a novel PET radiotracer specific to M2-activated microglia has been developed and tested in preclinical models [187], which has the potential to refine treatments promoting anti-inflammatory effects of M2 microglia. However, MRI techniques, including DWI, fMRI, and MRS, have not yet been shown to have utility in timing treatments of TBI. Furthermore, MRI has been best utilized in mTBI outside of the acute phase of injury [174], while the physiologic insights possible with fMRI and/or MRS would likely be most useful in the acute phase of moderate and severe TBI. Multimodal imaging, including MRI, is difficult to perform in this population due to the practical difficulties of transporting a critically ill patient out of the intensive care unit for long imaging procedures, or placing an intubated and mechanically ventilated patient into the MRI scanner [188]. This critical patient population is thus generally unable to benefit from multimodal imaging, and prognostication is typically done using non-multimodal imaging,, such as CT scans [189,190].

Several groups have trialed methods to reduce the limitations associated with multimodal imaging as an adjunct to the treatment of TBI. Portable imaging devices have been developed that can be brought to the patient’s bedside, removing the logistical challenges and risk associated with transferring a critically ill or injured patient [191]. Portable low-field MRI has shown potential to refine treatment in the neurocritical care setting [192], however, the resulting images are significantly lower in quality as compared to a traditional MRI, which limits clinical utility. Another method to reduce the limitations of multimodal imaging is the creation of longitudinal prospective multi-institutional studies to standardize imaging techniques and link acute imaging results with long-term functional outcomes. One such example is the TRACK-TBI study, which has leveraged this model to uncover early multimodal imaging markers in mTBI that predict long-term outcomes [193]. Furthermore, the data generated by studies like TRACK-TBI can be used to train, validate, and test machine learning models to uncover prognostic insights into TBI physiology that could be used to guide treatment. Various forms of machine learning models have been used to generate prognostic insights into TBI based on multimodal imaging data, including convolutional neural networks [194] and linear support vector machine analysis [195]. It has been suggested that prognostic information derived from machine learning models may help guide clinical treatment decisions, including appropriateness of aggressive therapies [196].

In its current form, MMM consists of collecting and deriving a wide range of cerebral physiologic data, including cerebral physical, hemodynamic, metabolic, and electrophysiologic parameters, which are collected and output as a continuous stream of high-frequency data points [121]. This real-time tracking of cerebral physiology has great potential to guide multimodal treatment of secondary injury following TBI. Certain parameters (ICP, P_bt_O_2_) have reliably shown to be effective in guiding treatment of moderate and severely injured TBI patients [117,197]. Ischemia and metabolic disruption in TBI lead to abnormal and pathologic electrical events, including cortical spreading depolarizations (CSDs) which are common after TBI and known to play a role in secondary injury [72,198,199,200]. Reliable means of identifying and targeting CSDs in TBI has been proposed to be a major need in the current landscape of TBI treatment [198,201,202], thus placing electrophysiologic MMM techniques in a valuable role for the guidance of therapeutic interventions targeting mechanisms underlying secondary injury in TBI. While no work to date has identified a multimodal treatment for CSDs in TBI, a recent paper showed a benefit from ketamine in multiple preclinical models of CSD [203], which in conjunction with the successes of ketamine as a multimodal treatment [23], suggests it may have potential for the multimodal treatment of CSDs.

However, MMM has a major operational limitation in that modern MMM generates such high volumes of data that drawing treatment-guiding conclusions is challenging for clinicians [188]. Additionally, there is a lack of clinical guidelines for the interpretation of the majority of this data, with the most recent Brain Trauma Foundation guidelines including recommendations only on multimodal measurement of ICP, CPP, and S_jv_O_2_-based arteriovenous oxygen content difference due to their known potential for guiding treatment decisions [10]. Thus, modern MMM produces a high volume of data on multiple physiologic parameters, without a large degree of guidance for clinicians caring for patients with TBI. Several recent groups have advocated for the use of machine learning and “big data” analytic approaches to synthesize this information and classify patients into distinct physiologic states, allowing for an individualized yet systematic approach to treating the evolving physiologic derangements caused by TBI [13,204,205]. For example, while the BOOST-II trial was not statistically powered to guide outcomes-oriented treatment, a machine learning analysis of BOOST-II data used a combination of logistic regression, elastic net, and random forest machine learning methods to derive clinically applicable predictive models for ICP and brain oxygenation that could be used for early intervention and treatment of intracranial hypertension and hypoxia [206]. Moving forward, future work linking large high-fidelity data sets of MMM-derived physiologic data with long-term clinical outcomes could be used to further drive advances in TBI treatment [188,207]. For example, the Targeted Evaluation, Action, and Monitoring group integrates a large volume of clinical monitoring data to categorize patients by disease phenotype and deploy a targeted treatment plan based on their clinical status, with the ability to reassess and determine the need for future treatments [13]. Another novel approach is to integrate TBI serum biomarkers with information from multimodal imaging and MMM to create an individualized “-omics” data set (proteomics, metabolomics, physiomics, etc.), which through extensive machine learning analysis, can lead to development of a comprehensive physiology-based therapeutic plan [188].

MMM-guided treatment for TBI is a promising component of a multimodal treatment paradigm but has several limitations that may affect implementation. It has been argued that the parameters measured in MMM may only reflect micro-scale physiology in a particular brain region, not the CNS as a whole [208]. A combination of intraparenchymal sensors has shown promise in better characterizing the general cerebral environment and local hemodynamics, however, these results must be validated with more established techniques [209]. While the data derived from MMM can effectively guide treatment [113,121], the invasiveness of MMM probes has limited MMM to use in moderate and severe TBI. Invasive MMM is generally considered safe [210], however, noninvasive approaches to MMM could expand its use to the mTBI population. Noninvasive measurements of ICP, including optic nerve sheath diameter, have been shown to correlate with invasive ICP measurement techniques and help guide surgical management of TBI [112,211], and thus may allow MMM to guide treatment in mTBI without insertion of invasive monitoring devices. However, noninvasive techniques have significant variation in inter-user reliability [212], and have not demonstrated an ability to monitor changes in ICP over time [213], which significantly limits their ability to guide treatment.

### 4.4. Limitations

This review is not without limitations. While our primary literature review was extensive, encompassing more than 4000 primary sources, the heterogeneity of the TBI literature and the wide variety of research methods used make it possible that some articles may have been missed by our search. The studies found included a wide variety of injury severity (mild to severe) and time after injury (acute, subacute to chronic). Additionally, many multimodal monitoring and imaging techniques can give insight into the ongoing secondary insults following TBI (cerebral edema, elevated ICP, etc.), but do not necessarily measure changes in underlying pathophysiology, which at times must be inferred. Lastly, while multimodal imaging, neuromonitoring, and therapeutic interventions are well studied in TBI, an operational definition is not always available for all studies. What exactly is or is not “multimodal” thus has a degree of variability that makes comparisons between studies more challenging. Our definitions of multimodal diagnostic and therapeutic techniques are based upon a synthesis of the available literature, but other groups may find other operational definitions more useful.

## 5. Conclusions

TBI is one of the most common global causes of death and disability, with a relative lack of effective treatments. As a disease with heterogeneous pathophysiology, including ischemia, oxidative stress, neuroinflammation, and excitotoxicity, monomodal treatment paradigms targeting a single aspect of TBI pathophysiology have shown poor results in clinical trials. Despite initial excitement, combination therapies targeting multiple individual mechanisms simultaneously have to date also fallen short of expectations, although there have been relatively few such studies. A multimodal treatment paradigm integrating physiologic information derived in part from imaging and neuromonitoring can be used to form a pathophysiologic assessment, which may then be treated with targeted pharmacologic and/or non-pharmacologic interventions. Multimodal interventions, both pharmacologic and non-pharmacologic, have shown great promise in preclinical studies but have yet to undergo widespread testing in clinical trials. Imaging and neuromonitoring have demonstrated the ability to help guide multimodal treatment of TBI, due to insights into prognosis and disease pathophysiology. Future interventions, including multi-institutional outcome-driven data sets, refinement of machine learning models, and increasing the study of female TBI pathophysiology, represent promising areas of ongoing development in order to fully implement the promise of multi-mechanistic therapeutic approaches.

## Figures and Tables

**Figure 1 jcm-12-02179-f001:**
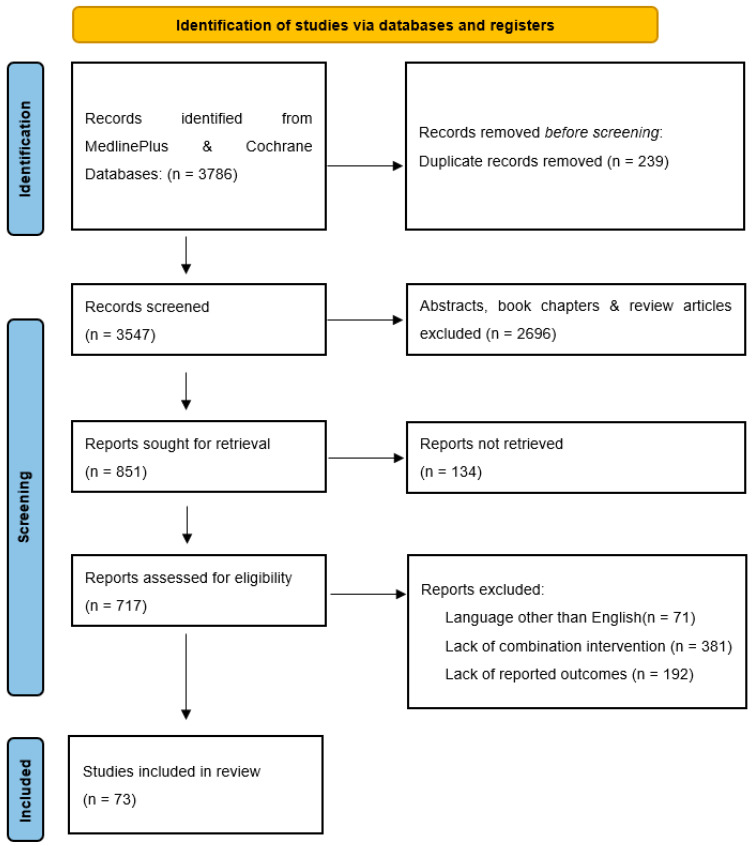
Flow diagram of literature search, article screening, full-text review, and exclusion criteria based on PRISMA reporting guidelines [11].

**Table 1 jcm-12-02179-t001:** Pharmacologic interventions trialed for TBI treatment.

Model	Population	Sample Size	%Fem	Intervention	Outcome of Combination Therapy	Reference
Clinical Trials						
Case Series	Severe Human TBI	2	0%	Cerebrolysin, citicoline	Improved long-term neurologic recovery with combined therapy in acute TBI	Trimmel, 2022 [15]
Retrospective Observational	Severe Human TBI	117	20.6%	HTS, mannitol, barbiturates, propofol, fentanyl	Propofol and fentanyl reduced ICP, but less than HTS in patients with acute TBI	Colton, 2014 [16]
Randomized Placebo Controlled	Severe Human TBI	14	21.4%	Probenecid, NAC	No harmful effects from combo in Phase I trial in acute pediatric TBI	Clark, 2017 [17]
Preclinical Trials						
CCI	Mouse	36	0%	NSC, olfactory ensheathing cells, valproic acid	Improved behavioral function and NSC neuronal differentiation	Liu, 2022 [18]
CCI	Mouse	292	45.9%	Apocynin, salubrinal, TBHQ	Improved functional outcomes, brain lesion development, and reduced inflammation	Davis, 2022 [19]
Biopsy Punch	Rat	45	0%	NSC, curcumin nanoparticles	Reduced glial activation and edema, improved recovery	Narouiepour, 2022 [20]
CHI	Mouse	26	0%	Minocycline, NAC	Improved memory function and reduced neuronal loss	Whitney, 2021 [21]
CCI	Mouse	10	0%	Apocynin, TBHQ	Reduced white matter disruption	Chandran, 2021 [22]
Weight Drop	Mouse	44	0%	Ketamine, perampanel	Reduced neurobehavioral dysfunction and NF-kB/iNOS expression	Alqahtani, 2020 [23]
Weight Drop	Rat	32	0%	Felbamate, levetiracetam	Reduced pro-inflammatory cytokines and histologic damage	Bayhan, 2020 [24]
Weight Drop	Rat	42	0%	Doxycycline, tocopherol	Reduced neurobehavioral deficits, ROS, and pro-inflammatory cytokines	Rana, 2020 [25]
Weight Drop	Rat	30	0%	MDL28170, BMSC	Reduced inflammation and improved survival of stem cells, with reduced neurobehavioral impairment	Hu, 2019 [26]
Weight Drop	Rat	24	100%	Lomerizine, YM872, Brilliant Blue G	Decreased microglial activation and myelin disruption without affecting neurobehavioral impairment	Mao, 2018 [27]
CCI	Mouse	48	0%	GSNO, CAPE	Reduced oxidative stress and mitochondrial dysfunction, improved neurobehavioral function	Khan, 2018 [28]
CCI	Mouse	NR	0%	Minocycline, NAC	Prevented loss of oligodendrocytes following CCI	Sangobowale, 2018 [29]
CCI	Rat	24	0%	Magnesium, NAT	Reduced BBB disruption and improved functional outcomes	Ameliorate, 2017 [30]
CCI	Mouse	31	0%	DHA, NSC	Improved neurogenesis and functional outcomes, with increased astrocyte and microglia activation	Ghazale, 2018 [31]
CCI	Mouse	NR	0%	Apocynin, TBHQ	Improved function and lesion volume	Chandran, 2018 [32]
mCCI	Rat	NR	0%	Minocycline, NAC	Protected oligodendrocytes and increased M1/M2 microglial activation	Haber, 2018 [33]
FPI	Rat	70	0%	Memantine, estradiol	Improved functional deficits and reduced neuronal degeneration	Day, 2017 [34]
CCI	Rat	96	0%	Magnesium, PEG	Improved CNS penetration of magnesium, increased neuroprotection	Busingye, 2016 [35]
CCI	Rat	207	0%	BMSC, propranolol	Decreased long-term neurobehavioral deficits	Kota, 2016 [36]
CHI	Rat	35	0%	Carnosine, Cyclosporine	Decreased pro-inflammatory cytokines and neuronal apoptosis	Baky, 2016 [37]
In-Vitro TBI	Rat Brain Slices	NR	0%	Memantine, estradiol	Reduced neuronal death	Lamprecht, 2015 [38]
Cryo-Injury	Mouse	70	0%	BMDC, lipoic acid	Increased cell growth in perilesional penumbra, decreased astrocyte infiltration, increased microglial activation	Paradells, 2015 [39]
CCI	Rat	40	0%	Progesterone, vitamin D	Reduced neuronal loss and astrocyte activation, mediated through downregulations in TLR4/NF-kB	Tang, 2015 [40]
CCI	Rat	31	0%	G-CSF, hUCB	Reduced activation of microglia and improved neurogenesis and functional recovery	Acosta, 2014 [41]
CCI	Rat	40	0%	Etanercept, lithium	Reduced edema and neuronal/glial apoptosis	Ekici, 2014 [42]
mCCI	Rat	NR	NR	Minocycline, NAC	Reduced neuroinflammation and neurobehavioral deficits	Haber, 2013 [43]
CCI	Mouse	126	0%	Lithium, valproic acid	Reduced BBB disruption, lesion volume, neuronal degeneration, and functional deficits	Yu, 2013 [44]
CCI	Rat	74	NR	Progesterone, magnesium	Reduced neuronal apoptosis and neurobehavioral deficits	Uysal, 2013 [45]
CCI	Mouse	50	0%	Melatonin, dexamethasone	Reduced lesion volume, oxidative stress, and functional deficits	Campolo, 2013 [46]
CCI	Mouse	44	0%	Dexamethasone, bortezomib	Reduced edema and BBB disruption	Thal, 2013 [47]
CCI	Rat	128	0%	Progesterone, vitamin D	Reduced neuronal loss and astrocyte activation	Tang, 2013 [48]
CCI	Rat	38	0%	Nimodipine, melatonin	Worsened edema and neuronal necrosis compared to melatonin alone	Ismailoglu, 2012 [49]
CCI	Rat	46	0%	Progesterone, vitamin D	Improved neurobehavioral function and increased astrocyte activation	Hua, 2012 [50]
CHI	Mouse	39	0%	VEGF, FGF2	Improved functional outcomes, no additional benefit versus monotherapy	Thau-Zuchman, 2012 [51]
CHI	Rat	35	100%	Estrogen, progesterone	Less reduction of edema and anti-inflammatory cytokines versus estrogen alone	Khaksari, 2011 [52]
CCI	Rat	50	0%	Minocycline, melatonin	No significant effect	Kelso, 2012 [53]
CHI	Rat	32	0%	Magnesium, MK801	Reduced edema and BBB disruption, but no greater effect than monotherapy	Imer, 2009 [54]
CCI	Rat	NR	NR	L-arginine, D-arginine, SOD, catalase	Increased nitric oxide and cerebral blood flow after TBI	Cherian, 2003 [55]
mCCI	Rat	30	0%	MK801, scopolamine	Improved hippocampal neuronal death and associated memory deficits	Jenkins, 1999 [56]
FPI	Rat	42	NR	Morphine, scopolamine	Improved functional outcomes	Lyeth, 1993 [57]

BBB, blood-brain barrier; BMDC, bone marrow derived cells; BMSC, bone marrow derived mesenchymal stem cells; CAPE, caffeic acid phenethyl ester; CCI, controlled cortical impact; CHI, closed head injury; DHA, docosahexaenoic acid; FGF2, fibroblast growth factor 2; G-CSF, granulocyte colony stimulating factor; HTS, hypertonic saline; GSNO, S-nitrosoglutathione; hUCB, human umbilical cord blood; ICP, intracranial pressure; mCCI, mild controlled cortical impact; NAC, N-acetylcysteine; NAT, N-acetyl-L-tryptophan; NR, not reported; NSC, neural stem cell; PEG, polyethylene glycol; ROS, reactive oxygen species; SOD, superoxide dismutase; TBHQ, tert-butylhydroquinone; TBI, traumatic brain injury; VEGF, vascular endothelial growth factor.

**Table 2 jcm-12-02179-t002:** Multimodal non-pharmacologic methods used in the treatment of TBI.

Model	Population	Sample Size	%Fem	Intervention	Outcome of Combination Therapy	Reference
Clinical Trials						
Randomized Prospective	Mild- Moderate Human TBI	166	44%	Cognitive training + rTMS vs. Cognitive training alone	Improved neurologic and functional outcomes in chronic TBI rehabilitation	Zhou, 2021 [84]
Case Report	Severe Human TBI	1	0%	rTMS + Neuromotor training	Improved motor function in chronic TBI rehabilitation	Martino Cinnera, 2016 [85]
Preclinical Trials						
CCI	Rat	97	0%	TMS + EE vs. TMS alone	Improved motor and sensory function	Shin, 2018 [86]
FPI	Rat	46	0%	EE + MEOS vs. EE alone	Improved neurocognitive dysfunction	Maegele, 2005 [87]
FPI	Rat	24	0%	EE + MEOS vs. EE alone	Improved neurocognitive dysfunction, reduced neuronal apoptosis and astrocyte activation	Maegele, 2005 [88]

EE, environmental enrichment; MEOS, multimodal early onset stimulation; rTMS, repetitive transcranial magnetic stimulation; TBI, traumatic brain injury; TMS, transcranial magnetic stimulation.

**Table 4 jcm-12-02179-t004:** Multimodal neuromonitoring studied in the setting of TBI.

TBI Severity	Study Design	Sample Size	%Fem	Neuromonitoring Parameters	Outcome of MMM-Guided Treatment	Reference
Clinical Trials						
Moderate- Severe	Retrospective Observational	61	29.5%	ICP, CPP, PRx	May predict need for long-term treatment of seizures after TBI	Appavu, 2022 [108]
Severe	Retrospective Cohort	49	20.4%	ICP, P_bt_O_2_, CPP	Improved treatment of cerebral hypoxia and hypertension without improvement in long-term outcomes	Lang, 2022 [109]
Severe	Retrospective Observational	20	15%	ICP, CPP, P_bt_O_2_, LPR	Enabled diagnosis and treatment of cerebral metabolic crisis	Marini, 2022 [110]
Moderate- Severe	Prospective Interventional	5	100%	ICP, P_bt_O_2_, LPR	Improved cerebral metabolic dysfunction	Khellaf, 2022 [111]
Severe	Case Report	1	0%	ICP, P_bt_O_2_, CPP, PRx,	Guided need for surgical intervention	Robinson, 2021 [112]
Moderate- Severe	Retrospective Observational	85	31.8%	ICP, CPP, PRx	Helped guide clinical treatment in pediatric TBI, reduced length of time on mechanical ventilation	Appavu, 2021 [113]
Severe	Retrospective Observational	81	19.8%	ICP, CPP, PRx	Improved clinical outcomes	Petkus, 2020 [114]
Moderate- Severe	Retrospective Observational	38	32%	ICP, CPP, P_bt_O_2_, PaO_2_	Characterized etiology of cerebral hypoxemia and guided treatment	Dellazizzo, 2018 [115]
Severe	Prospective Randomized Cohort	119	21%	ICP, P_bt_O_2_	Reduced hypoxia with trend toward better outcomes than ICP alone	Okonkwo, 2017 [116]
Moderate- Severe	Retrospective Cohort	30	10%	ICP, CPP	Reduced mortality and length of ICU stay	Luca, 2015 [117]
Severe	Prospective Observational	18	38.9%	ICP, CPP	Reduced mortality and improved long-term neurologic outcomes	Dunham, 2006 [118]
Severe	Prospective Randomized Cohort	82	15.9%	ICP, CPP, P_bt_O_2_, CBF, pH, SvjO_2_, PRx	Treatment guided by MMM associated with improved outcomes	Isa, 2003 [119]

CBF, cerebral blood flow; CPP, cerebral perfusion pressure; ICP, intracranial pressure; ICU, intensive care unit; LPR, lactate-pyruvate ratio; MMM, multimodal neuromonitoring; PaO2, arterial blood oxygen partial pressure; PbtO2, brain tissue oxygenation; PRx, cerebrovascular pressure reactivity index; SvjO2, jugular venous oxygen saturation.

## Data Availability

Not applicable.

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
