# Peer review of "Multi-Mechanistic Approaches to the Treatment of Traumatic Brain Injury: A Review"

_jcm, 2023, doi:10.3390/jcm12062179_

Round 1

Reviewer 1 Report

In this manuscript, the authors summarized the combination of therapeutic interventions for TBI, in both clinical and/or pre-clinical studies in recent 15 years. In addition, the authors discussed the current and ongoing work relating to multimodal treatment, with an emphasis on the integration of nonpharmacologic treatments and endogenous mechanisms. This information will be helpful for clinicians and researchers to fully implement the multi-mechanistic therapeutic approaches for TBI.

There are a couple of minor suggestions:

1.       Page 6, line 143: “enzymes” was misspelled.

2.       In the discussion section, the authors can include the iron chelator as a potential treatment for TBI. There are many studies revealed a connection between acute CNS injuries and ferroptosis. Inhibit ferroptosis by iron chelators, such as Deferoxamine, may have neuroprotective effects after acute CNS injuries.

Author Response

In this manuscript, the authors summarized the combination of therapeutic interventions for TBI, in both clinical and/or pre-clinical studies in recent 15 years. In addition, the authors discussed the current and ongoing work relating to multimodal treatment, with an emphasis on the integration of nonpharmacologic treatments and endogenous mechanisms. This information will be helpful for clinicians and researchers to fully implement the multi-mechanistic therapeutic approaches for TBI.

There are a couple of minor suggestions:

1. Page 6, line 143: “enzymes” was misspelled.

 This typo has been corrected. (Page 6, line 155).

2. In the discussion section, the authors can include the iron chelator as a potential treatment for TBI. There are many studies revealed a connection between acute CNS injuries and ferroptosis. Inhibit ferroptosis by iron chelators, such as Deferoxamine, may have neuroprotective effects after acute CNS injuries.

A discussion of ferroptosis as a key mediator of oxidative stress induced damage, as well as the treatments targeting ferroptosis including deferoxamine, has been added to the discussion section (Page 14, lines 443-451).

Reviewer 2 Report

This comprehensive review on traumatic brain injury treatment perspectives is well designed and structured, and it provides evidence-based conclusions outlining future directions both for preclinical and clinical research in the field of study. While the main body of the text is solid, there are several points where it can be improved and made less ambiguous. Finally, a careful proof-read is needed to eliminate syntaxes.

Specific points

1. P 6, L134-146. While evidence for involvement of ROC in TBI pathology is overwhelming, literature on tissue pO2 in hyperacute trauma is limited, yet it should be cited in the review. This as it is counterintuitive that ‘hypoxia/ischaemia’ prevails meaning low/no tissue O2, yet O2 is required for generation of ‘reactive oxygen species’.

2. P6, L174-19. One should not overlook release of Ca2+ from internal stores as sources of increased intracellular Ca2+

3. P9, L287. ‘cerebral metabolomics …by MRS’. This is awkward jargon, as in vivo MRS detects cerebral metabolites, their tissue concentrations are commonly given either as relative or absolute means. These have little or nothing to do with ‘metabolomics’. Revise

4. P13, L 373. The first sentence is redundant, the statement has been used several times earlier.

5. P15, L470-7. Writing about contusion/peri-contusion tissue can be condensed to give the essence of their characteristics.

6. P16, L 560. ‘multimodal imaging’ and little later ‘multimodal MRI’. This is confusing, as multimodal is also used in the context of ‘multimodal neuromonitoring’. While it is fine to use ‘multimodal imaging’ followed by a definition what is meant by it, i.e. use of CT, MRI, PET, NIRS, EEG etc, but ‘multimodal MRI’ remain an oddity. MRI field uses ‘multimodal’, however, by doing so it is defined what that includes. I would propose the authors abandon ‘multimodal MRI’ and replace that by ‘structural MRI, functional MRI, diffusion MRI, MRS etc.

7. P17, L 579. ‘charactering cerebral metabolites’. This is yet another awkward jargon around MRS and certainly untrue. As stated above, MRS allows to quantify cerebral metabolites, but to characterise cerebral metabolites is not done.

8. P 17, L601. ‘low-filed multimodal MRI’, ‘multimodal in the context of low-field MRI is an overkill, remove.

9. P18, L617 ’MMM’ is defined earlier.

10. P18, L634. Techniques used in MMM are defined earlier, P11.

11. P19, Conclusions. Use of ‘multimodal’ in several contexts is confusing, see notes above.

Author Response

This comprehensive review on traumatic brain injury treatment perspectives is well designed and structured, and it provides evidence-based conclusions outlining future directions both for preclinical and clinical research in the field of study. While the main body of the text is solid, there are several points where it can be improved and made less ambiguous. Finally, a careful proof-read is needed to eliminate syntaxes.

Specific points

1. P 6, L134-146. While evidence for involvement of ROC in TBI pathology is overwhelming, literature on tissue pO2 in hyperacute trauma is limited, yet it should be cited in the review. This as it is counterintuitive that ‘hypoxia/ischaemia’ prevails meaning low/no tissue O2, yet O2 is required for generation of ‘reactive oxygen species’.

The relationship between hyperacute tissue hypoxemia, metabolic dysfunction and ROS generation has been addressed, including the counterintuitive influence of tissue hypoxia on ROS generation (Page 6, lines 150-152).

2. P6, L174-19. One should not overlook release of Ca2+ from internal stores as sources of increased intracellular Ca2+

This has been added as a source of [Ca2+]I (Page 7, lines 189-190).

3. P9, L287. ‘cerebral metabolomics …by MRS’. This is awkward jargon, as in vivo MRS detects cerebral metabolites, their tissue concentrations are commonly given either as relative or absolute means. These have little or nothing to do with ‘metabolomics’. Revise

Revised wording to clarify measurement of ratios indicative of metabolic changes (Page 9, lines 303-305).

4. P13, L 373. The first sentence is redundant, the statement has been used several times earlier.

Redundant information on TBI morbidity and mortality removed (Page 13, lines 402-404).

5. P15, L470-7. Writing about contusion/peri-contusion tissue can be condensed to give the essence of their characteristics.

Discussion of the contused and peri-contused tissue has been condensed to the most relevant characteristics of each region (Pages 15-16, lines 507-516).

6. P16, L 560. ‘multimodal imaging’ and little later ‘multimodal MRI’. This is confusing, as multimodal is also used in the context of ‘multimodal neuromonitoring’. While it is fine to use ‘multimodal imaging’ followed by a definition what is meant by it, i.e. use of CT, MRI, PET, NIRS, EEG etc, but ‘multimodal MRI’ remain an oddity. MRI field uses ‘multimodal’, however, by doing so it is defined what that includes. I would propose the authors abandon ‘multimodal MRI’ and replace that by ‘structural MRI, functional MRI, diffusion MRI, MRS etc.

Revised paragraph on multimodal imaging to discuss specific multimodal imaging techniques, and replaced “multimodal MRI” with either discussion of MRI in general or specific MRI techniques/sequences when appropriate (Pages 17-18, lines 606-644).

7. P17, L 579. ‘charactering cerebral metabolites’. This is yet another awkward jargon around MRS and certainly untrue. As stated above, MRS allows to quantify cerebral metabolites, but to characterise cerebral metabolites is not done.

Description of MRS revised for increased accuracy and clarity (Page 17, lines 624-626).

8. P 17, L601. ‘low-filed multimodal MRI’, ‘multimodal in the context of low-field MRI is an overkill, remove.

Low field portable MRI no longer described as “multimodal” (Page 18, line 649).

9. P18, L617 ’MMM’ is defined earlier.

Redundant definition of MMM removed (Page 18, line 664).

10. P18, L634. Techniques used in MMM are defined earlier, P11.

Redundant description of MMM techniques removed (Page 18, lines 681-682).

11. P19, Conclusions. Use of ‘multimodal’ in several contexts is confusing, see notes above.

For the sake of clarity, removed use of “multimodal” in contexts other than multimodal treatment (Page 20, lines 750 & 754).
